# Study on Infinitely Many Solutions for a Class of Fredholm Fractional Integro-Differential System

**Dongping Li** [1], **Yankai Li** [2,*] **and Fangqi Chen** [3,4]

1   Department of Mathematics, Xi'an Technological University, Xi'an 710021, China
2   School of Automation and Information Engineering, Xi'an University of Technology, Xi'an 710048, China
3   Department of Mathematics, Nanjing University of Aeronautics and Astronautics, Nanjing 211106, China
4   College of Mathematics and Systems Science, Shandong University of Science and Technology, Qingdao 266590, China
*   Correspondence: liyankai@xaut.edu.cn

**Abstract:** This paper deals with a class of nonlinear fractional Sturm–Liouville boundary value problems. Each sub equation in the system is a fractional partial equation including the second kinds of Fredholm integral equation and the $p$-Laplacian operator, simultaneously. Infinitely many solutions are derived due to perfect involvements of fractional calculus theory and variational methods with some simpler and more easily verified assumptions.

**Keywords:** fractional integro-differential equation; Sturm–Liouville boundary condition; variational method

**MSC:** 26A33; 34B15; 35A15

## 1. Introduction

Nano/microactuators, as an indispensable portion of nano/microelectromechanical systems, are always subject to different inherent nonlinear forces. Many studies show that an integro-differential equation is generated in the modeling process of the nano/microactuator governing equation owing to axial forces ([1–3]). In [4,5], the following nanoactuator beam equation augmented to boundary conditions and containing an integro-differential expression, was discussed

$$\begin{cases} \frac{d^4 f}{dt^4} - \left(\mu \int_0^1 \left(\frac{df}{dt}\right)^2 dt + L\right)\frac{d^2 f}{dt^2} + \frac{\theta}{f^\eta} + \frac{\kappa}{(r+f)^2} + \frac{s}{f} = 0, \ t \in [0,1], \\ f(0) = f(1) = 0, \ f'(0) = f'(1) = 0, \end{cases} \tag{1}$$

where $f$ and $t$ denote the deflection and length of the beam, respectively. $\mu$, $L$, $\kappa$ and $r$ denote some inherent nonlinear forces. Actually, in practical engineering applications, actuators are constructed by the billions for chipsets, therefore, developing more effective and accurate strategies for the study of nano/microactuator structures is of great significance.

Furthermore, it is often not appropriate to establish models with delayed behaviors by ordinary differential equations or partial differential equations, while integral equations are ideal tools. Moreover, fractional calculus operators are convolution operators (For details, please refer to the definitions of fractional integral and differential operators in [6], in which the definitions involving convolution integrals.), because they are nonlocal and have full-memory function, and those characteristics can be well used to describe various phenomena and complex processes involving delay and global correlations. For this reason, fractional calculus has been extensively applied in interdisciplinary fields such as fluid and viscoelastic mechanics, control theory, signal and image processing, electricity, physical, etc., (see [7–9]). Therefore, matching fractional calculus operators and integro-differential equations is ideal to complete the mathematical modeling of practical problems. Taking

into account the effect of a full-memory system, the integer derivatives in Equation (1) can be substituted for fractional ones. Inspired by this fact in Equation (1), Shivanian [10] introduced the following overdetermined Fredholm fractional integro-differential equations

$$
\begin{cases}
{}_tD_T^{\alpha_j}(a_j(t){}_0D_t^{\alpha_j}u_j(t)) = \lambda F_{u_j}(t, u_1(t), \ldots, u_m(t)) + \int_0^T k_j(t,s)u_j(s)ds, \ t \in (0,T), \ j = 1,2,\ldots,m, \\
u_j(t) = \int_0^T k_j(t,s)u_j(s)ds, \ t \in (0,T), \ j = 1,2,\ldots,m, \\
u_j(0) = u_j(T) = 0, \ j = 1,2,\ldots,m,
\end{cases}
\tag{2}
$$

where $\alpha_j \in (0,1]$, $a_j(t) \in L^\infty[0,T]$, $j = 1,2,\ldots,m$. The existence of at least three weak solutions was obtained through the three critical points theorem.

　　Committed to fully considering more general systems, this paper studies a class of nonlinear Fredholm fractional integro-differential equations with $p$-Laplacian operator and Sturm–Liouville boundary conditions as below

$$
\begin{cases}
{}_tD_T^{\gamma_j}(k_j(t)\Phi_p({}_0^C D_t^{\gamma_j}z_j(t))) + l_j(t)\Phi_p(z_j(t)) \\
\quad = \lambda f_{z_j}(t, z_1(t), \ldots, z_m(t)) + \int_0^T g_j(t,s)\Phi_p(z_j(s))ds, \ t \in [0,T], \ j = 1,2,\ldots,m, \\
z_j(t) = \int_0^T g_j(t,s)\Phi_p(z_j(s))ds, \ t \in [0,T], \ j = 1,2,\ldots,m, \\
c_jk_j(0)\Phi_p(z_j(0)) - c'_j{}_tD_T^{\gamma_j-1}(k_j(0)\Phi_p({}_0^C D_t^{\gamma_j}z_j(0))) = 0, \ j = 1,2,\ldots,m, \\
d_jk_j(T)\Phi_p(z_j(T)) + d'_j{}_tD_T^{\gamma_j-1}(k_j(T)\Phi_p({}_0^C D_t^{\gamma_j}z_j(T))) = 0, \ j = 1,2,\ldots,m,
\end{cases}
\tag{3}
$$

where $c_j, c'_j, d_j$ and $d'_j$ are positive constants, $\lambda \in (0, +\infty)$ is a parameter, $k_j, l_j \in L^\infty[0,T]$ with $\widehat{k_j} = ess\inf_{[0,T]} k_j(t) > 0$ and $\widehat{l_j} = ess\inf_{[0,T]} l_j(t) \geq 0$, $j = 1,2,\ldots,m$. For $1 < p < \infty$, $\Phi_p(s) = |s|^{p-2}s(s \neq 0)$, $\Phi_p(0) = 0$, $f : [0,T] \times \mathbb{R}^m \to \mathbb{R}$ satisfies $f(\cdot, z_1(t), \ldots, z_m(t)) \in C[0,T]$ and $f(t, \cdot, \ldots, \cdot) \in C^1[\mathbb{R}^m]$, $g_j(\cdot, \cdot) \in C([0,T], [0,T])$. ${}_0^C D_t^{\gamma_j}$ and ${}_tD_T^{\gamma_j}$ denote the left Caputo fractional derivative and right Riemann–Liouville fractional derivative of order $\gamma_j$, respectively, which are defined by Kilbas et al. in [6]

$$
{}_tD_T^{\gamma_j}u(t) = (-1)^n \frac{d^n}{dt^n}{}_tD_T^{\gamma_j-n}u(t) = \frac{(-1)^n}{\Gamma(n-\gamma_j)}\frac{d^n}{dt^n}\int_t^T (\zeta - t)^{n-\gamma_j-1}u(\zeta)d\zeta,
\tag{4}
$$

$$
{}_0^C D_t^{\gamma_j}u(t) = {}_0D_t^{\gamma_j-n}u^{(n)}(t) = \frac{1}{\Gamma(n-\gamma_j)}\int_0^t (t - \zeta)^{n-\gamma_j-1}u^{(n)}(\zeta)d\zeta,
\tag{5}
$$

for $\forall u(t) \in AC([0,T], \mathbb{R})$, $n - 1 \leq \gamma_j < n$, $n \in \mathbb{N}$.

　　We emphasize that this paper extends previous results in several directions, which are listed as follows: (i) In recent years, a large number of existence results for fractional differential equations have been acquired by variational methods and critical point theory ([11–14]). However, not many research works are available in related references to handle fractional integro-differential equations, let alone involving the $p$-Laplacian operator and Sturm–Liouville boundary conditions. (ii) It is not hard to see that Equation (3) can turn into the Dirichlet boundary value problem Equation (2) under $p = 2, c'_j = d'_j = 0, l_j(t) \equiv 0, j = 1,2,\ldots,m$, which means that Equation (2) is a special case of Equation (3). Furthermore, since the $p$-Laplacian operator is considered with $1 < p < \infty$ in the paper, the linear differential operator ${}_tD_{T0}^{\gamma C}D_t^{\gamma}$ is extended to the nonlinear differential operator ${}_tD_T^{\gamma}\Phi_p({}_0^C D_t^{\gamma})$. In short, the form of Equation (3) is more generalized, as well as the boundary value conditions. (iii) Infinitely many solutions are obtained in this paper with some simpler and more easily verified assumptions. Hence, our work improves and replenishes some existing results form the literature.

## 2. Preliminaries

　　Assume $H$ is a Banach space and $\mathcal{F} \in C^1(H, \mathbb{R})$. Functional $\mathcal{F}$ satisfies the Palais–Smale condition if each sequence $\{z_k\}_{k=1}^\infty \subset H$ such that $\{\mathcal{F}(z_k)\}$ is bounded and $\lim_{k\to\infty} \mathcal{F}'(z_k) = 0$ possesses strongly convergent subsequence in $H$.

**Theorem 1** ([15]). *Let $H$ be an infinite-dimensional Banach space, $\mathcal{F} \in C^1(H, \mathbb{R})$ is an even functional and satisfies the Palais–Smale condition. Assume that:*

(i) $\mathcal{F}(0) = 0$. *There exist $\tau > 0$ and $\eta > 0$ such that $\overline{Y}_\tau \subset \{z \in H \mid \mathcal{F}(z) \geq 0\}$ and $\mathcal{F}(z) \geq \eta$ for all $z \in \partial Y_\tau$, where $Y_\tau = \{z \in H \mid \|z\| < \tau\}$;*

(ii) *For any finite dimensional subspace $H_0 \subset H$, the set $H_0 \bigcap \{z \in H \mid \mathcal{F}(z) \geq 0\}$ is bounded.*

*Then, $\mathcal{F}$ has infinitely many critical points.*

**Definition 1.** *Let $1 < p < \infty$, $\frac{1}{p} < \gamma_j \leq 1$, $j = 1, 2, \ldots, m$. Define the fractional derivative space $H = \Pi_{j=1}^{j=m} H^{\gamma_j, p}$ with the weighted norm*

$$\|Z\|_H = \sum_{j=1}^{j=m} \|z_j\|_{(\gamma_j, p)}, \ z_j \in H^{\gamma_j, p}, \ Z = (z_1, \ldots, z_m) \in H, \tag{6}$$

*where*

$$H^{\gamma_j, p} = \{z_j \in AC([0, T], \mathbb{R}) : {}_0^C D_t^{\gamma_j} z_j(t) \in L^p([0, T], \mathbb{R})\}$$

*as the closure of $C^\infty([0, T], \mathbb{R})$ endowed with the norm*

$$\|z_j\|_{(\gamma_j, p)} := \left( \int_0^T | z_j(t) |^p \, dt + \int_0^T | {}_0^C D_t^{\gamma_j} z_j(t) |^p \, dt \right)^{\frac{1}{p}}, \forall z_j \in H^{\gamma_j, p}. \tag{7}$$

*$H^{\gamma_j, p}$ is a reflexive and separable Banach space [16]. Therefore, $H$ also is a reflexive and separable Banach space.*

**Lemma 1** ([13]). *For any $z_j(t) \in H^{\gamma_j, p}$, $1 < p, q < \infty$ with $\frac{1}{p} + \frac{1}{q} = 1$, there exists a constant $W_{(\gamma_j, p)} = \max \left\{ \frac{T^{\gamma_j - \frac{1}{p}}}{\Gamma(\gamma_j)((\gamma_j - 1)q + 1)^{\frac{1}{q}}}, 1 \right\} + \left[ \frac{2^{p-1}}{T} \max \left\{ 1, \left( \frac{T^{\gamma_j}}{\Gamma(\gamma_j + 1)} \right)^p \right\} \right]^{\frac{1}{p}}$ such that $\|z_j\|_\infty \leq W_{(\gamma_j, p)} \|z_j\|_{(\gamma_j, p)}$, $j = 1, 2, \ldots, m$.*

Taking into account Lemma 1, one has

$$\|z_j\|_\infty \leq \frac{W_{(\gamma_j, p)}}{(\min\{\widehat{k}_j, \widehat{l}_j\})^{\frac{1}{p}}} \left( \int_0^T l_j(t) | z_j(t) |^p \, dt + \int_0^T k_j(t) | {}_0^C D_t^{\gamma_j} z_j(t) |^p \, dt \right)^{\frac{1}{p}}, \ \forall z_j(t) \in H^{\gamma_j, p}, \tag{8}$$

$j = 1, 2, \ldots, m$. In order to describe it more easily for the further analysis, denote

$$W_j = \frac{W_{(\gamma_j, p)}}{(\min\{\widehat{k}_j, \widehat{l}_j\})^{\frac{1}{p}}}, \ \widehat{W} = \max_{1 \leq j \leq m} \{W_j\}. \tag{9}$$

Obviously, the norm defined by (7) is equivalent to

$$\|z_j\|_{(\gamma_j, p)} = \left( \int_0^T l_j(t) | z_j(t) |^p \, dt + \int_0^T k_j(t) | {}_0^C D_t^{\gamma_j} z_j(t) |^p \, dt \right)^{\frac{1}{p}}, \ j = 1, 2, \ldots, m. \tag{10}$$

We work with the norm (10) hereinafter.

**Lemma 2** ([17]). *Let $1 < p < \infty$, $\gamma_j \in (\frac{1}{p}, 1]$, $j = 1, 2, \ldots, m$. Suppose that any sequence $\{z_{k, j}\}$ converges to $z_j$ in $H^{\gamma_j, p}$ weakly. Then, $z_{k, j} \rightarrow z_j$ in $C([0, T])$ as $k \rightarrow \infty$.*

**Lemma 3** ([18]). *Let $H_j$ be any finite-dimensional subspace of $H^{\gamma_j, p}$, $j = 1, 2, \ldots, m$. There exists a constant $\zeta_0 > 0$ such that $\text{meas}\{t \in [0, T] :| z_j(t) |\geq \zeta_0 \|z_j\|_{(\gamma_j, p)}\} \geq \zeta_0$, $\forall z_j(t) \in H_j \setminus \{0\}$.*

**Lemma 4** ([6]). *Let $\gamma > 0$, $p \geq 1$, $q \geq 1$ and $\frac{1}{p} + \frac{1}{q} \leq 1 + \gamma$ ($p \neq 1, q \neq 1$ in the case when $\frac{1}{p} + \frac{1}{q} = 1 + \gamma$). If $z_1 \in L^p([a,b])$ and $z_2 \in L^q([a,b])$, then, $\int_a^b ({}_aD_t^{-\gamma}z_1(t))z_2(t)dt = \int_a^b z_1(t)({}_tD_b^{-\gamma}z_2(t))dt$.*

**Lemma 5.** *It is said $Z = (z_1, \ldots, z_m) \in H$ is a weak solution of Equations (3), if the following equation holds*

$$\sum_{j=1}^m \left\{ \int_0^T k_j(t)\Phi_p({}_0^C D_t^{\gamma_j} z_j(t)){}_0^C D_t^{\gamma_j} y_j(t) + l_j(t)\Phi_p(z_j(t))y_j(t)dt + \frac{c_j}{c_j'}k_j(0)\Phi_p(z_j(0))y_j(0) + \frac{d_j}{d_j'}k_j(T)\Phi_p(z_j(T))y_j(T) \right\}$$

$$= \sum_{j=1}^m \left\{ \int_0^T \int_0^T g_j(t,s)\Phi_p(z_j(s))y_j(t)dsdt + \lambda \int_0^T f_{z_j}(t, z_1(t), \ldots, z_m(t))y_j(t)dt \right\}, \forall Y = (y_1, \ldots, y_m) \in H. \tag{11}$$

**Proof.** Consider (4) and (5), the boundary conditions in Equation (3) and Lemma 4 yield:

$$\int_0^T {}_tD_T^{\gamma_j}(k_j(t)\Phi_p({}_0^C D_t^{\gamma_j} z_j(t)))y_j(t)dt$$

$$= -\int_0^T y_j(t)d[{}_tD_T^{\gamma_j-1}(k_j(t)\Phi_p({}_0^C D_t^{\gamma_j} z_j(t)))]$$

$$= {}_tD_T^{\gamma_j-1}\left(k_j(0)\Phi_p({}_0^C D_t^{\gamma_j} z_j(0))\right)y_j(0) - {}_tD_T^{\gamma_j-1}\left(k_j(T)\Phi_p({}_0^C D_t^{\gamma_j} z_j(T))\right)y_j(T) + \int_0^T {}_tD_T^{\gamma_j-1}(k_j(t)\Phi_p({}_0^C D_t^{\gamma_j} z_j(t)))y_j'(t)dt \tag{12}$$

$$= \frac{c_j}{c_j'}k_j(0)\Phi_p(z_j(0))y_j(0) + \frac{d_j}{d_j'}k_j(T)\Phi_p(z_j(T))y_j(T) + \int_0^T k_j(t)\Phi_p({}_0^C D_t^{\gamma_j} z_j(t)){}_0^C D_t^{\gamma_j} y_j(t)dt.$$

Substituting $y_j(t)$ into Equation (3) and integrating on both sides from 0 to $T$, then summing from $j = 1$ to $j = m$ and combining with (12), we can obtain Equation (11). The proof is completed. $\square$

**Remark 1.** *For any $z_j \in H^{\gamma_j,p} \subset C([0,T])$, $j = 1, 2, \ldots, m$, from Equation (3) we have*

$${}_tD_T^{\gamma_j}(k_j(t)\Phi_p({}_0^C D_t^{\gamma_j} z_j(t))) + l_j(t)\Phi_p(z_j(t)) = \lambda f_{z_j}(t, z_1(t), \ldots, z_m(t)) + \int_0^T g_j(t,s)\Phi_p(z_j(s))ds, t \in [0,T],$$

*because $f(t, \cdot, \ldots, \cdot) \in C^1[R^m]$, $z_j(t) = \int_0^T g_j(t,s)\Phi_p(z_j(s))ds \in H^{\gamma_j,p}$ and*

$${}_tD_T^{\gamma_j}(k_j(t)\Phi_p({}_0^C D_t^{\gamma_j} z_j(t))) = \left({}_tD_T^{\gamma_j-1}(k_j(t)\Phi_p({}_0^C D_t^{\gamma_j} z_j(t)))\right)',$$

*one gets*

$${}_tD_T^{\gamma_j-1}(k_j(t)\Phi_p({}_0^C D_t^{\gamma_j} z_j(t))) \in AC([0,T]).$$

*Hence, the terms ${}_tD_T^{\gamma_j-1}(k_j(0)\Phi_p({}_0^C D_t^{\gamma_j} z_j(0)))$ and ${}_tD_T^{\gamma_j-1}(k_j(T)\Phi_p({}_0^C D_t^{\gamma_j} z_j(T)))$ exist in this paper.*

Consider the functional $\mathcal{F} : H \to \mathbb{R}$ with

$$\mathcal{F}(Z) := \frac{1}{p} \sum_{j=1}^{j=m} \int_0^T k_j(t) \mid {}_0^C D_t^{\gamma_j} z_j(t) \mid^p + l_j(t) \mid z_j(t) \mid^p dt + \sum_{j=1}^{j=m} \left[ \frac{c_j}{pc_j'} k_j(0) \mid z_j(0) \mid^p + \frac{d_j}{pd_j'} k_j(T) \mid z_j(T) \mid^p \right]$$

$$- \sum_{j=1}^{j=m} \int_0^T G_j(z_j(t)) dt - \lambda \int_0^T f(t, z_1(t), \ldots, z_m(t)) dt$$

$$= \frac{1}{p} \sum_{j=1}^{j=m} \|z_j\|_{(\gamma_j, p)}^p + \sum_{j=1}^{j=m} \left[ \frac{c_j}{pc_j'} k_j(0) \mid z_j(0) \mid^p + \frac{d_j}{pd_j'} k_j(T) \mid z_j(T) \mid^p \right]$$

$$- \sum_{j=1}^{j=m} \int_0^T G_j(z_j(t)) dt - \lambda \int_0^T f(t, z_1(t), \ldots, z_m(t)) dt, \tag{13}$$

where $G_j(z_j(t)) = \frac{1}{2} \int_0^T g_j(t,s) \Phi_p(z_j(s)) z_j(t) ds, t \in (0, T), j = 1, 2, \ldots, m$. Owing to $z_j(t) = \int_0^T g_j(t,s) \Phi_p(z_j(s)) ds, j = 1, 2, \ldots, m$, the Gâteaux derivative of $G_j$ is

$$G_j'(z_j)(y_j) = \lim_{h \to 0} \frac{G_j(z_j + hy_j) - G_j(z_j)}{h} \tag{14}$$

$$= \lim_{h \to 0} \frac{\frac{1}{2} \int_0^T g_j(t,s) \Phi_p(z_j(s) + hy_j(s))(z_j(t) + hy_j(t)) - g_j(t,s) \Phi_p(z_j(s)) z_j(t) ds}{h}$$

$$= \lim_{h \to 0} \frac{\frac{1}{2} h^2 y_j^2(t) + hz_j(t) y_j(t)}{h} = z_j(t) y_j(t) = \int_0^T g_j(t,s) \Phi_p(z_j(s)) y_j(t) ds, j = 1, 2, \ldots, m.$$

Then, combining the continuity of $f$ and (14), we can see that $\mathcal{F} \in C^1(H, \mathbb{R})$ and

$$\mathcal{F}'(Z)(Y) = \sum_{j=1}^{j=m} \left\{ \int_0^T k_j(t) \Phi_p({}_0^C D_t^{\gamma_j} z_j(t)) {}_0^C D_t^{\gamma_j} y_j(t) + l_j(t) \Phi_p(z_j(t)) y_j(t) dt + \frac{c_j}{c_j'} k_j(0) \Phi_p(z_j(0)) y_j(0) \right. \tag{15}$$

$$\left. + \frac{d_j}{d_j'} k_j(T) \Phi_p(z_j(T)) y_j(T) - \int_0^T \int_0^T g_j(t,s) \Phi_p(z_j(s)) y_j(t) ds dt - \lambda \int_0^T f_{z_j}(t, Z(t)) y_j(t) dt \right\}, \forall Z, Y \in H.$$

Notice that, the critical point of $\mathcal{F}$ is the weak solution of Equation (3).

## 3. Main Results

First, some hypotheses related to nonlinearity $f$ are given, which play important roles in the remaining discussion.

$(H_0)$ $\lim\limits_{\forall j: |z_j| \to \infty} \frac{f(t, Z(t))}{\sum_{j=1}^{j=m} |z_j|^p} = \infty$ uniformly for $t \in [0, T], Z(t) = (z_1(t), \ldots, z_m(t)) \in \mathbb{R}^m$;

$(H_1)$ $0 \le f(t, Z(t)) = o(\sum_{j=1}^{j=m} |z_j|^p)$ as $\sum_{j=1}^{j=m} |z_j| \to 0$ uniformly for $t \in [0, T]$;

$(H_2)$ For any $Z(t) = (z_1(t), \ldots, z_m(t)) \in \mathbb{R}^m, f(t, Z(t)) = \sum_{j=1}^{j=m} \frac{\eta_j}{p} |z_j|^p - J(t, Z(t))$ with $J(t, 0) \equiv 0$, and

$$\min_{1 \le j \le m} \{\eta_j\} > \frac{1}{\lambda \zeta_0^{p+1}} (\frac{3}{2} + p \sum_{j=1}^{j=m} [\frac{c_j}{pc_j'} k_j(0) + \frac{d_j}{pd_j'} k_j(T)] W_j^p),$$

$$\sum_{j=1}^{j=m} (\frac{\eta_j}{p} + \frac{\beta_j}{2\lambda}) |z_j|^{\omega_j} \le J(t, Z(t)) \le \sum_{j=1}^{j=m} \delta_j |z_j|^{\omega_j},$$

where $\omega_j \in (0, p), \delta_j > 0, \zeta_0 > 0$ is a constant and $\widehat{\beta}$ is introduced thereinafter, $j = 1, 2, \ldots, m$.

**Lemma 6.** $\mathcal{F}$ *satisfies the Palais–Smale condition under* $(H_0)$.

**Proof.** Suppose that sequence $\{\mathcal{F}(Z_k)\}_{k \in \mathbb{N}}$ is bounded and $\lim\limits_{k \to \infty} \mathcal{F}'(Z_k) = 0, Z_k(t) = (z_{k,1}(t), \ldots, z_{k,m}(t))$. We claim that $\{Z_k\}_{k \in \mathbb{N}}$ is bounded in $H$. Indeed, assume

$\forall j : \|z_{k,j}\|_{(\gamma_j,p)} \to \infty(k \to \infty)$. From $(H_0)$, for any $L > 0$, there exists $k_0 \in \mathbb{N}$ such that

$$\frac{f(t, Z_k(t))}{\sum_{j=1}^{j=m} \|z_{k,j}\|_{(\gamma_j,p)}^p} \geq L, \; \forall \, k > k_0, \; t \in [0, T]. \tag{16}$$

For any fixed $k_* \in \mathbb{N}$ with $k_* > k_0$, from the integral mean value theorem, there exists $\xi(k_*) \in (0, 1]$ such that

$$\int_0^T f(t, Z_{k_*}(t))dt = Tf(\xi(k_*)T, Z_{k_*}(\xi(k_*)T)). \tag{17}$$

Combining (16) and (17) yields

$$\frac{\int_0^T f(t, Z_{k_*}(t))dt}{\sum_{j=1}^{j=m} \|z_{k_*,j}\|_{(\gamma_j,p)}^p} = \frac{Tf(\xi(k_*)T, Z_{k_*}(\xi(k_*)T))}{\sum_{j=1}^{j=m} \|z_{k_*,j}\|_{(\gamma_j,p)}^p} \geq \frac{TL \sum_{j=1}^{j=m} \|z_{k_*,j}\|_{(\gamma_j,p)}^p}{\sum_{j=1}^{j=m} \|z_{k_*,j}\|_{(\gamma_j,p)}^p} = TL.$$

Hence, we can get

$$\frac{\int_0^T f(t, Z_k(t))dt}{\sum_{j=1}^{j=m} \|z_{k,j}\|_{(\gamma_j,p)}^p} \geq TL, \; \forall \, k > k_0, \; t \in [0, T]. \tag{18}$$

In view of (8), (9), (13) and (18) we have

$$\frac{\mathcal{F}(Z_k(t))}{\sum_{j=1}^{j=m} \|z_{k,j}\|_{(\gamma_j,p)}^p} = \frac{1}{p} + \frac{\sum_{j=1}^{j=m} \left[ \frac{c_j}{pc_j'} k_j(0) \mid z_{k,j}(0) \mid^p + \frac{d_j}{pd_j'} k_j(T) \mid z_{k,j}(T) \mid^p \right]}{\sum_{j=1}^{j=m} \|z_{k,j}\|_{(\gamma_j,p)}^p}$$

$$- \frac{\sum_{j=1}^{j=m} \int_0^T G_j(z_{k,j}(t))dt + \lambda \int_0^T f(t, Z_k(t))dt}{\sum_{j=1}^{j=m} \|z_{k,j}\|_{(\gamma_j,p)}^p}$$

$$\leq \frac{1}{p} + \frac{\sum_{j=1}^{j=m} [\frac{c_j}{pc_j'} k_j(0) + \frac{d_j}{pd_j'} k_j(T)] W_j^p \|z_{k,j}\|_{(\gamma_j,p)}^p}{\sum_{j=1}^{j=m} \|z_{k,j}\|_{(\gamma_j,p)}^p} - \lambda TL$$

$$\leq \frac{1}{p} + \sum_{j=1}^{j=m} [\frac{c_j}{pc_j'} k_j(0) + \frac{d_j}{pd_j'} k_j(T)] W_j^p - \lambda TL. \tag{19}$$

Choose $L$ large enough such that $\frac{1}{p} + \sum_{j=1}^{j=m} [\frac{c_j}{pc_j'} k_j(0) + \frac{d_j}{pd_j'} k_j(T)] W_j^p - \lambda TL < -1$, then combining (19) yields that $\mathcal{F}(Z_k(t)) \leq -\sum_{j=1}^{j=m} \|z_{k,j}\|_{(\gamma_j,p)}^p$, which means that $\mathcal{F}(Z_k(t)) \to -\infty$ as $\|z_{k,j}\|_{(\gamma_j,p)} \to \infty, \forall j = 1, 2, \ldots, m$. It contradicts that $\{\mathcal{F}(Z_k)\}$ is bounded. Hence, $\{Z_k\}$ is bounded in $H$. Because of the reflexivity of $H$, we get that $Z_k \rightharpoonup Z^*$ in $H$ (up to subsequences). From Lemma 2, we have $Z_k \to Z^*$ uniformly in $C([0, T]^m)$ and $L^p([0, T]^m)$. Then,

$$\begin{cases} (\mathcal{F}'(Z_k) - \mathcal{F}'(Z^*))(Z_k - Z^*) \to 0, \; k \to \infty, \\ \int_0^T (f_{z_j}(t, Z_k(t)) - f_{z_j}(t, Z^*(t)))(z_{k,j}(t) - z_j^*(t))dt \to 0, \; k \to \infty, , j = 1, 2, \ldots, m, \\ \int_0^T \mid z_{k,j}(t) - z_j^*(t) \mid^2 dt \to 0, z_{k,j}(0) - z_j^*(0) \to 0, z_{k,j}(T)) - z_j^*(T) \to 0, \; k \to \infty, j = 1, 2, \ldots, m. \end{cases} \tag{20}$$

From (15), we obtain that

$$(\mathcal{F}'(Z_k) - \mathcal{F}'(Z^*))(Z_k - Z^*) = \mathcal{F}'(Z_k)(Z_k - Z^*) - \mathcal{F}'(Z^*)(Z_k - Z^*)$$

$$= \sum_{j=1}^{j=m} \left\{ \int_0^T k_j(t) \left( \Phi_p(^C_0 D_t^{\gamma_j} z_{k,j}(t)) - \Phi_p(^C_0 D_t^{\gamma_j} z_j^*(t)) \right) ^C_0 D_t^{\gamma_j}(z_{k,j}(t) - z_j^*(t)) + l_j(t) \left( \Phi_p(z_{k,j}(t)) - \Phi_p(z_j^*(t)) \right) (z_{k,j}(t) - z_j^*(t)) dt \right. \quad (21)$$

$$- \int_0^T \int_0^T g_j(t,s) \left( \Phi_p(z_{k,j}(s)) - \Phi_p(z_j^*(s)) \right) (z_{k,j}(t) - z_j^*(t)) ds dt + \frac{c_j}{c_j'} k_j(0) \left( \Phi_p(z_{k,j}(0)) - \Phi_p(z_j^*(0)) \right) (z_{k,j}(0) - z_j^*(0))$$

$$+ \frac{d_j}{d_j'} k_j(T) \left( \Phi_p(z_{k,j}(T)) - \Phi_p(z_j^*(T)) \right) (z_{k,j}(T) - z_j^*(T)) - \lambda \int_0^T (f_{z_j}(t, Z_k(t)) - f_{z_j}(t, Z^*(t)))(z_{k,j}(t) - z_j^*(t)) dt \right\};$$

moreover,

$$\int_0^T \int_0^T g_j(t,s) \left( \Phi_p(z_{k,j}(s)) - \Phi_p(z_j^*(s)) \right) (z_{k,j}(t) - z_j^*(t)) ds dt = \int_0^T |z_{k,j}(t) - z_j^*(t)|^2 dt. \quad (22)$$

Denote

$$\Psi_{k,j}(\gamma_j, p) = \int_0^T k_j(t) \left( \Phi_p(^C_0 D_t^{\gamma_j} z_{k,j}(t)) - \Phi_p(^C_0 D_t^{\gamma_j} z_j^*(t)) \right) ^C_0 D_t^{\gamma_j}(z_{k,j}(t) - z_j^*(t)) dt,$$

$$\Psi_{k,j}(p) = \int_0^T l_j(t) \left( \Phi_p(z_{k,j}(t)) - \Phi_p(z_j^*(t)) \right) (z_{k,j}(t) - z_j^*(t)) dt,$$

combining (20), (21) and (22), we obtain $\sum_{j=1}^{j=m} \{\Psi_{k,j}(\gamma_j, p) + \Psi_{k,j}(p)\} \to 0$ as $k \to \infty$. As in the discussion of $\Theta(\alpha, p), \Theta(p)$ in [19], we can get

$$\Psi_{k,j}(\gamma_j, p) + \Psi_{k,j}(p) \geq \begin{cases} e_j \|z_{k,j} - z_j^*\|_{\gamma_j, p}^p, & p \geq 2, \\ e_j' \|z_{k,j} - z_j^*\|_{(\gamma_j, p)}^2 (\|z_{k,j}\|_{L^p}^p + \|z_j^*\|_{L^p}^p)^{\frac{p-2}{p}}, & 1 < p < 2, \end{cases}$$

where $e_j, e_j'$ are constants, $j = 1, 2, \ldots, m$. Based on the above discussion, we can obtain $\|z_{k,j} - z_j^*\|_{(\gamma_j, p)} \to 0, j = 1, 2, \ldots, m$, for all $1 < p < \infty$. Hence, the Palais–Smale condition holds. □

**Theorem 2.** *Assume that $(H_0)$ and $(H_1)$ hold and $f(t, Z) = f(t, -Z)$. Then, Equation (3) has infinitely many solutions with $\frac{1}{Tp\widehat{W}^p} - \widehat{\beta} > 0$ and $0 < \lambda < \infty$.*

**Proof.** Due to $f(t, Z) = f(t, -Z)$, it is easy to verify that $\mathcal{F}$ is even. Obviously, $\mathcal{F}(0) = 0$. Taking into account $(H_1)$ that, for any $\varepsilon > 0$, there exists $r(\varepsilon)$ such that

$$f(t, Z(t)) \leq \varepsilon \sum_{j=1}^{j=m} |z_j|^p, \forall t \in [0, T], \sum_{j=1}^{j=m} |z_j| \leq r(\varepsilon). \quad (23)$$

Further, $g_j(\cdot, \cdot) \in C([0, T], [0, T])$ means that the kernel $g_j$ is bounded by, say $\beta_j$, i.e., $|g_j(t, s)| \leq \beta_j$, and

$$G_j(z_j(t)) = \frac{1}{2} \int_0^T g_j(t,s) \Phi_p(z_j(s)) z_j(t) ds \leq \frac{\beta_j}{2} z_j(t) \|z_j\|_\infty^{p-1} \leq \frac{\beta_j}{2} \|z_j\|_\infty^p, j = 1, 2, \ldots, m. \quad (24)$$

Let $\tau = \frac{r}{\widehat{W}}$. For any $Z \in \overline{Y}_\tau$, one has $\|Z\|_H = \sum_{j=1}^{j=m} \|z_j\|_{(\gamma_j, p)} \leq \frac{r}{\widehat{W}}$. Then,

$$\frac{r}{\widehat{W}} \geq \sum_{j=1}^{j=m} \|z_j\|_{(\gamma_j, p)} \geq \sum_{j=1}^{j=m} \frac{1}{W_j} \|z_j\|_\infty \geq \frac{1}{\widehat{W}} \sum_{j=1}^{j=m} \|z_j\|_\infty, \quad (25)$$

which means that $\sum_{j=1}^{j=m} \|z_j\|_\infty \leq r(\varepsilon)$. At this point, from (13), (23) and (24) we can see

$$
\begin{aligned}
\mathcal{F}(Z) &\geq \frac{1}{p}\sum_{j=1}^{j=m}\|z_j\|_{(\gamma_j,p)}^p - \sum_{j=1}^{j=m}\int_0^T \frac{\beta_j}{2}\|z_j\|_\infty^p\, dt - \lambda\int_0^T \varepsilon\sum_{j=1}^{j=m}|z_j|^p\, dt \\
&\geq \frac{1}{p}\sum_{j=1}^{j=m}\|z_j\|_{(\gamma_j,p)}^p - \sum_{j=1}^{j=m}(\frac{T\beta_j}{2} + \lambda\varepsilon T)W_j^p\|z_j\|_{(\gamma_j,p)}^p \\
&\geq [\frac{1}{p} - (\frac{T\widehat{\beta}}{2} + \lambda\varepsilon T)\widehat{W}^p]\frac{1}{m^p}\left(\sum_{j=1}^{j=m}\|z_j\|_{(\gamma_j,p)}\right)^p \\
&= [\frac{1}{p} - (\frac{T\widehat{\beta}}{2} + \lambda\varepsilon T)\widehat{W}^p]\frac{1}{m^p}\|Z\|_H^p,\ \forall Z\in\overline{Y}_\tau,
\end{aligned}
\tag{26}
$$

where $\widehat{\beta} = \max\limits_{1\leq j\leq m}\{\beta_j\}$. Choose $\varepsilon = \frac{1}{2\lambda}(\frac{1}{Tp\widehat{W}^p} - \widehat{\beta})$, from (26), we get

$$
\mathcal{F}(Z) \geq \frac{1}{2pm^p}\|Z\|_H^p \geq 0.
\tag{27}
$$

Hence, $\overline{Y}_\tau \subset \{Z\in H\mid \mathcal{F}(Z)\geq 0\}$ and $\mathcal{F}(Z) \geq \frac{1}{2pm^p}\|Z\|_H^p, \forall Z\in\partial Y_\tau$. Therefore, the condition $(i)$ in Theorem 1 holds.

For any finite-dimensional space $H_0\subset H$, we claim that $\widetilde{H} = H_0\bigcap\{Z\in H\mid \mathcal{F}(Z)\geq 0\}$ is bounded. Assume that there exists at least a sequence $\{Z_k\}\subset\widetilde{H}$ such that $\|Z_k\|_H\to\infty$ as $k\to\infty$. From $\mathcal{F}(Z_k)\geq 0$ and (19), we obtain

$$
0 \leq \frac{\mathcal{F}(Z_k(t))}{\sum_{j=1}^{j=m}\|z_{k,j}\|_{(\gamma_j,p)}^p} \leq \frac{1}{p} + \sum_{j=1}^{j=m}[\frac{c_j}{pc_j'}k_j(0) + \frac{d_j}{pd_j'}k_j(T)]W_j^p - \lambda TL.
$$

Since $L$ is arbitrary, we draw a contradiction. Therefore, $\widetilde{H} = H_0\bigcap\{Z\in H\mid \mathcal{F}(Z)\geq 0\}$ is bounded. Based on Theorem 1, functional $\mathcal{F}$ has infinitely many critical points, which means that Equation (3) has infinitely many solutions in $H$. $\square$

**Theorem 3.** *Assume that* $(H_2)$ *holds and* $J(t, Z) = J(t, -Z)$. *Then, Equation (3) has infinitely many solutions with* $\sum_{j=1}^{j=m}\frac{1}{p} - \left(\frac{\beta_j T}{2} + \frac{\lambda T\eta_j}{p}\right)W_j^p > 0$.

**Proof.** Suppose that the sequence $\{\mathcal{F}(Z_k)\}_{k\in\mathrm{N}}$ is bounded and $\lim\limits_{k\to\infty}\mathcal{F}'(Z_k) = 0$, $Z_k(t) = (z_{k,1}(t),\ldots,z_{k,m}(t))$. In what follows, we prove that $\mathcal{F}$ satisfies the Palais–Smale condition. Indeed, assume $\forall j : \|z_{k,j}\|_{(\gamma_j,p)}\to\infty (k\to\infty)$, from (13), (24), $(H_2)$ and (8), we have

$$
\begin{aligned}
\frac{1}{p}\sum_{j=1}^{j=m}\|z_{k,j}\|_{(\gamma_j,p)}^p &\leq \mathcal{F}(Z_k) + \sum_{j=1}^{j=m}\int_0^T G_j(z_j(t))dt + \lambda\int_0^T f(t,z_1(t),\ldots,z_m(t))dt \\
&\leq \mathcal{F}(Z_k) + \sum_{j=1}^{j=m}\int_0^T \frac{\beta_j}{2}|z_{k,j}|^p\, dt + \lambda\sum_{j=1}^{j=m}\int_0^T \frac{\eta_j}{p}|z_{k,j}|^p - (\frac{\eta_j}{p} + \frac{\beta_j}{2\lambda})|z_{k,j}|^{\omega_j}\, dt \\
&\leq \mathcal{F}(Z_k) + \sum_{j=1}^{j=m}\left(\frac{\beta_j T}{2} + \frac{\lambda T\eta_j}{p}\right)W_j^p\|z_{k,j}\|_{(\gamma_j,p)}^p + \lambda T\sum_{j=1}^{j=m}(\frac{\eta_j}{p} + \frac{\beta_j}{2\lambda})W_j^{\omega_j}\|z_{k,j}\|_{(\gamma_j,p)}^{\omega_j},
\end{aligned}
\tag{28}
$$

namely

$$
\sum_{j=1}^{j=m}\left[\frac{1}{p} - \left(\frac{\beta_j T}{2} + \frac{\lambda T\eta_j}{p}\right)W_j^p\right]\|z_{k,j}\|_{(\gamma_j,p)}^p - \lambda T\sum_{j=1}^{j=m}(\frac{\eta_j}{p} + \frac{\beta_j}{2\lambda})W_j^{\omega_j}\|z_{k,j}\|_{(\gamma_j,p)}^{\omega_j} \leq \mathcal{F}(Z_k).
\tag{29}
$$

Recall that $\sum_{j=1}^{j=m} \frac{1}{p} - \left( \frac{\beta_j T}{2} + \frac{\lambda T \eta_j}{p} \right) W_j^p > 0$, $\omega_j \in (0, p)$ and $\{\mathcal{F}(Z_k)\}$ is bounded, we get a contradiction. Hence, $\{Z_k\}$ is bounded on $H$. The rest of the proof for the Palais–Smale condition is similar to that of Lemma 6, so we do not repeat it.

Let $\tau' \in (0, \frac{1}{W})$. For any $Z \in \overline{Y}_{\tau'}$, one has $\|Z\|_H = \sum_{j=1}^{j=m} \|z_j\|_{(\gamma_j, p)} \leq \tau' < \frac{1}{W}$. A similar analysis with (25) yields $\sum_{j=1}^{j=m} \|z_j\|_\infty < 1$. From (28), we get

$$
\begin{aligned}
\mathcal{F}(Z) &\geq \frac{1}{p} \sum_{j=1}^{j=m} \|z_j\|_{(\gamma_j, p)}^p - \sum_{j=1}^{j=m} \int_0^T \frac{\beta_j}{2} \mid z_j \mid^p dt - \lambda \sum_{j=1}^{j=m} \int_0^T \frac{\eta_j}{p} \mid z_j \mid^p - (\frac{\eta_j}{p} + \frac{\beta_j}{2\lambda}) \mid z_j \mid^{\omega_j} dt \\
&= \frac{1}{p} \sum_{j=1}^{j=m} \|z_j\|_{(\gamma_j, p)}^p - \sum_{j=1}^{j=m} \left[ \int_0^T \frac{\beta_j}{2} \mid z_j \mid^p dt + \lambda \int_0^T \frac{\eta_j}{p} \mid z_j \mid^p - (\frac{\eta_j}{p} + \frac{\beta_j}{2\lambda}) \mid z_j \mid^{\omega_j} dt \right] \\
&= \frac{1}{p} \sum_{j=1}^{j=m} \|z_j\|_{(\gamma_j, p)}^p - \sum_{j=1}^{j=m} \left[ \int_0^T (\frac{\beta_j}{2} + \frac{\lambda \eta_j}{p}) \mid z_j \mid^p - (\frac{\lambda \eta_j}{p} + \frac{\beta_j}{2}) \mid z_j \mid^{\omega_j} dt \right] \\
&= \frac{1}{p} \sum_{j=1}^{j=m} \|z_j\|_{(\gamma_j, p)}^p + \sum_{j=1}^{j=m} \int_0^T (\frac{\lambda \eta_j}{p} + \frac{\beta_j}{2}) \mid z_j \mid^{\omega_j} - (\frac{\lambda \eta_j}{p} + \frac{\beta_j}{2}) \mid z_j \mid^p dt \\
&\geq \frac{1}{p} \sum_{j=1}^{j=m} \|z_j\|_{(\gamma_j, p)}^p + \sum_{j=1}^{j=m} \int_0^T (\frac{\lambda \eta_j}{p} + \frac{\beta_j}{2}) \mid z_j \mid^p - (\frac{\lambda \eta_j}{p} + \frac{\beta_j}{2}) \mid z_j \mid^p dt \\
&= \frac{1}{p} \sum_{j=1}^{j=m} \|z_j\|_{(\gamma_j, p)}^p \geq \frac{1}{pm^p} (\sum_{j=1}^{j=m} \|z_j\|_{(\gamma_j, p)})^p = \frac{1}{pm^p} \|Z\|_H^p \geq 0, \ \forall \ Z \in \overline{Y}_{\tau'}.
\end{aligned}
$$

Clearly, $\overline{Y}_{\tau'} \subset \{Z \in H \mid \mathcal{F}(Z) \geq 0\}$ and $\mathcal{F}(Z) \geq \frac{1}{pm^p} \|Z\|_H^p, \forall Z \in \partial Y_{\tau'}$.

For any finite-dimensional space $H_0' \subset H$, we claim that $\widehat{H} = H_0' \bigcap \{Z \in H \mid \mathcal{F}(Z) \geq 0\}$ is bounded. Assume that there exists at least a sequence $\{Z_k\} \subset \widehat{H}$ such that $\|Z_k\|_H \to \infty$ as $k \to \infty$. Then, according to (19), $(H_2)$ and Lemma 3 we obtain

$$
\begin{aligned}
0 \leq \frac{\mathcal{F}(Z_k(t))}{\sum_{j=1}^{j=m} \|z_{k,j}\|_{(\gamma_j, p)}^p} &\leq \frac{1}{p} + \sum_{j=1}^{j=m} [\frac{c_j}{pc_j'} k_j(0) + \frac{d_j}{pd_j'} k_j(T)] W_j^p - \frac{\lambda \int_0^T \sum_{j=1}^{j=m} \frac{\eta_j}{p} \mid z_{k,j} \mid^p - J(t, Z_k(t)) dt}{\sum_{j=1}^{j=m} \|z_{k,j}\|_{(\gamma_j, p)}^p} \\
&\leq \frac{1}{p} + \sum_{j=1}^{j=m} [\frac{c_j}{pc_j'} k_j(0) + \frac{d_j}{pd_j'} k_j(T)] W_j^p - \frac{\sum_{j=1}^{j=m} \frac{\lambda \eta_j}{p} \int_{\Omega_{z_{k,j}}} \zeta_0^p \|z_{k,j}\|_{(\gamma_j, p)}^p dt}{\sum_{j=1}^{j=m} \|z_{k,j}\|_{(\gamma_j, p)}^p} + \frac{\lambda \int_0^T \sum_{j=1}^{j=m} \delta_j \mid z_{k,j} \mid^{\omega_j} dt}{\sum_{j=1}^{j=m} \|z_{k,j}\|_{(\gamma_j, p)}^p} \\
&\leq \frac{1}{p} + \sum_{j=1}^{j=m} [\frac{c_j}{pc_j'} k_j(0) + \frac{d_j}{pd_j'} k_j(T)] W_j^p - \frac{\lambda \zeta_0^{p+1}}{p} \min_{1 \leq j \leq m} \{\eta_j\} + \frac{\lambda T \sum_{j=1}^{j=m} \delta_j W_j^{\omega_j} \|z_{k,j}\|_{(\gamma_j, p)}^{\omega_j}}{\sum_{j=1}^{j=m} \|z_{k,j}\|_{(\gamma_j, p)}^p},
\end{aligned}
\tag{30}
$$

where $\Omega_{z_{k,j}} = \{t \in [0, T] : \mid z_{k,j}(t) \mid \geq \zeta_0 \|z_{k,j}\|_{(\gamma_j, p)}\}$ and $meas\{\Omega_{z_{k,j}}\} \geq \zeta_0$. Since $\min_{1 \leq j \leq m} \{\eta_j\} > \frac{1}{\lambda \zeta_0^{p+1}} (\frac{3}{2} + p \sum_{j=1}^{j=m} [\frac{c_j}{pc_j'} k_j(0) + \frac{d_j}{pd_j'} k_j(T)] W_j^p)$, then

$$
\frac{1}{p} + \sum_{j=1}^{j=m} [\frac{c_j}{pc_j'} k_j(0) + \frac{d_j}{pd_j'} k_j(T)] W_j^p - \frac{\lambda \zeta_0^{p+1}}{p} \min_{1 \leq j \leq m} \{\eta_j\} < -\frac{1}{2p},
\tag{31}
$$

based on $\omega_j \in (0, p)$ and $\|Z_k\|_H \to \infty$ as $k \to \infty$, we get

$$
\frac{\lambda T \sum_{j=1}^{j=m} \delta_j W_j^{\omega_j} \|z_{k,j}\|_{(\gamma_j, p)}^{\omega_j}}{\sum_{j=1}^{j=m} \|z_{k,j}\|_{(\gamma_j, p)}^p} \to 0, k \to \infty.
\tag{32}
$$

Combining (31) and (32), we obtain that $0 \leq \dfrac{\mathcal{F}(Z_k(t))}{\sum_{j=1}^{j=m} \|z_{k,j}\|_{(\gamma_j, p)}^p} < -\dfrac{1}{2p}$ as $k \to \infty$, which

draws a contradiction. Hence, $\widehat{H}$ is bounded. Based on Theorem 1, functional $\mathcal{F}$ has infinitely many critical points, which means that Equation (3) has infinitely many solutions in $H$. □

**Example 1.** *Focus on the following Fredholm fractional partial integro-differential equations with* $m = 3$ *and* $p = 4$:

$$
\begin{cases}
{}_tD_1^{0.5}((t+1)\Phi_4({}_0^CD_t^{0.5}z_1(t))) + (\frac{1}{2}+t)\Phi_4(z_1(t)) = D_{z_1}f(t,z_1(t),z_2(t),z_3(t)) + \int_0^1 10^{-5}t\sin(s)\Phi_4(z_1(s))ds, t \in [0,1], \\
z_1(t) = \int_0^1 10^{-5}t\sin(s)\Phi_4(z_1(s))ds, \ t \in [0,1], \\
{}_tD_1^{0.6}((t^2+1)\Phi_4({}_0^CD_t^{0.6}z_2(t))) + (\frac{1}{3}+t^2)\Phi_4(z_2(t)) = D_{z_2}f(t,z_1(t),z_2(t),z_3(t)) + \int_0^1 10^{-5}t^2\sin(s)\Phi_4(z_2(s))ds, t \in [0,1], \\
z_2(t) = \int_0^1 10^{-5}t^2\sin(s)\Phi_4(z_2(s))ds, \ t \in [0,1], \\
{}_tD_1^{0.75}((t^3+1)\Phi_4({}_0^CD_t^{0.75}z_3(t))) + (\frac{1}{4}+t^3)\Phi_4(z_3(t)) = D_{z_3}f(t,z_1(t),z_2(t),z_3(t)) + \int_0^1 10^{-5}t^3\sin(s)\Phi_4(z_3(s))ds, t \in [0,1], \\
z_3(t) = \int_0^1 10^{-5}t^3\sin(s)\Phi_4(z_3(s))ds, \ t \in [0,1], \\
\Phi_4(z_1(0)) - {}_tD_1^{-0.5}(\Phi_4({}_0^CD_t^{0.5}z_1(0))) = 0, \ \Phi_4(z_1(1)) + {}_tD_1^{-0.5}(\Phi_4({}_0^CD_t^{0.5}z_1(1))) = 0, \\
\Phi_4(z_2(0)) - {}_tD_1^{-0.4}(\Phi_4({}_0^CD_t^{0.6}z_2(0))) = 0, \ \Phi_4(z_2(1)) + {}_tD_1^{-0.4}(\Phi_4({}_0^CD_t^{0.6}z_2(1))) = 0, \\
\Phi_4(z_3(0)) - {}_tD_1^{-0.25}(\Phi_4({}_0^CD_t^{0.75}z_3(0))) = 0, \ \Phi_4(z_3(1)) + {}_tD_1^{-0.25}(\Phi_4({}_0^CD_t^{0.75}z_3(1))) = 0,
\end{cases} \tag{33}
$$

*where* $c_j = c_{j'} = 1, d_j = d_{j'} = \frac{1}{2}, j = 1,2,3$,

$$
f(t, z_1, z_2, z_3) = (1+t)
\begin{cases}
(z_1^4 + z_2^4 + z_3^4)^2, \ z_1^4 + z_2^4 + z_3^4 \leq 1, \\
2(z_1^4 + z_2^4 + z_3^4)^2 - (z_1^4 + z_2^4 + z_3^4)^{\frac{1}{2}}, \ z_1^4 + z_2^4 + z_3^4 > 1.
\end{cases}
$$

It is easy to verify that $f$ is continuous with respect to $t$ and continuously differentiable with respect to $z_1, z_2$ and $z_3$ (see Figures 1 and 2) and satisfies $(H_0)$ and $(H_1)$. Obviously, $k_1(0) = k_2(0) = k_3(0) = 1, k_1(1) = k_2(1) = k_3(1) = 2, \widehat{\beta} = 10^{-5}$. By direct calculation we have $\widehat{k}_1 = \widehat{k}_2 = \widehat{k}_3 = 1, \widehat{l}_1 = \frac{1}{2}, \widehat{l}_2 = \frac{1}{3}, \widehat{l}_3 = \frac{1}{4}$, and

$$
W_{(0.5,4)} = \max\left\{ \frac{1}{\Gamma(0.5)[(-\frac{1}{2})\frac{4}{3}+1]^{\frac{3}{4}}}, 1 \right\} + \left[ 8\max\left\{ 1, \left(\frac{1}{\Gamma(1.5)}\right)^4 \right\} \right]^{\frac{1}{4}} = 3.184,
$$

$$
W_{(0.6,4)} = \max\left\{ \frac{1}{\Gamma(0.6)[(-\frac{2}{5})\frac{4}{3}+1]^{\frac{3}{4}}}, 1 \right\} + \left[ 8\max\left\{ 1, \left(\frac{1}{\Gamma(1.6)}\right)^4 \right\} \right]^{\frac{1}{4}} = 3.072,
$$

$$
W_{(0.75,4)} = \max\left\{ \frac{1}{\Gamma(0.75)[(-\frac{1}{4})\frac{4}{3}+1]^{\frac{3}{4}}}, 1 \right\} + \left[ 8\max\left\{ 1, \left(\frac{1}{\Gamma(1.75)}\right)^4 \right\} \right]^{\frac{1}{4}} = 2.936,
$$

then

$$
\frac{W_{(0.5,4)}^4}{\min\{\widehat{k}_1, \widehat{l}_1\}} = 206, \quad \frac{W_{(0.6,4)}^4}{\min\{\widehat{k}_2, \widehat{l}_2\}} = 267, \quad \frac{W_{(0.75,4)}^4}{\min\{\widehat{k}_3, \widehat{l}_3\}} = 297,
$$

namely, $\widehat{W} = 297, \frac{1}{p\widehat{W}} = 8.4 \times 10^{-5}$, then $\frac{1}{p\widehat{W}} - \widehat{\beta} > 0$. Hence, from Theorem 2 we can see that Equation (33) has infinitely many solutions.

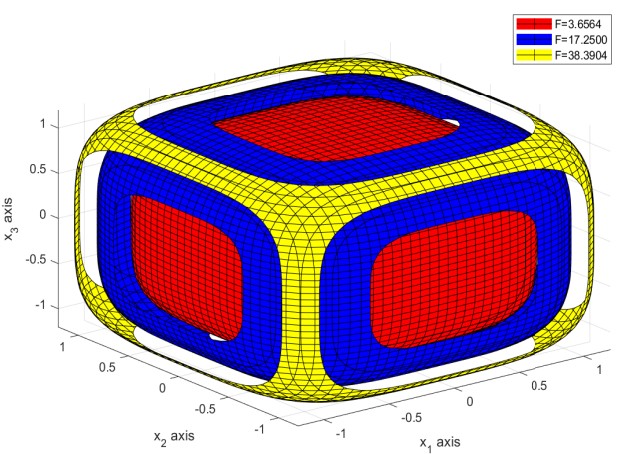

**Figure 1.** the contour-plot of Equation (33) for $t = 0$.

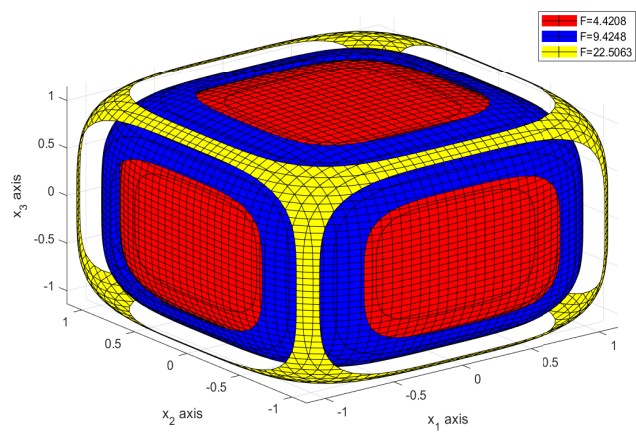

**Figure 2.** the contour-plot of Equation (33) for $t = 1$.

**Author Contributions:** Conceptualization, Y.L.; Investigation, D.L.; Writing—original draft, D.L.; Writing—review and editing, Y.L. and F.C. All authors have read and agreed to the published version of the manuscript.

**Funding:** This research was funded by National Natural Science Foundation of China grant numbers 12101481, 62103327, 11872201; Young Talent Fund of Association for Science and Technology in Shaanxi, China grant number 20220529; Young Talent Fund of Association for Science and Technology in Xi'an, China grant number 095920221344.

**Institutional Review Board Statement:** Not applicable.

**Informed Consent Statement:** Not applicable.

**Data Availability Statement:** Not applicable.

**Acknowledgments:** The authors would like to thank the editor and reviewers greatly for their precious comments and suggestions.

**Conflicts of Interest:** The authors declare no conflict of interest.

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
