# Peer review of "Study on Infinitely Many Solutions for a Class of Fredholm Fractional Integro-Differential System"

_fractalfract, doi:10.3390/fractalfract6090467_

Round 1

Reviewer 1 Report

In this paper the authors have studied a class of nonlinear fractional Sturm-Liouville boundary value problems. Also infinitely many solutions have been  derived by authors. The paper is a good study because of focusing on nonlinear and also fractional order. The introduction is good with enough background of this work. Ref [11] also is cited by authors which has been used as one of main references of this paper.  

The main conditions H0, H1 and H2 are presented and discussed very well. Also the mentioned example has been discussed with good results by authors. I propose "Accept in present form". 

Reviewer 2 Report

Report on "Study on innitely many solutions for a class of Fredholm fractional integro-dierential system" Author: Dongping Li , Yankai Li * , Fangqi Chen 

The paper is well written, with a relevant and useful provision of references. It is an extensively technical paper whose core is a sequence of theorems and proofs which are given throughout and associated with the analysis of Equation Eqs.(3), the fractional integro-dierential system subject to boundary conditions. The basic approach proposed is sound and well explained and the results are clearly presented. The focus of the paper, which is in regard to an analysis of the problem posed by Equation Eqs.(3), appears to be original and is the governing equation of a nano-actuator beam which including the eect of axial loads and dierent types of nonlinear forces. They prove the existence of innitely many solutions of the problem under weaker hypotheses by applying a variational method. The results are an improvement and generalization of the relative results obtained in the integer-order case. The important contributions that the paper provides are given in Section 3 with Theorem 3.1 and Theorem 3.2, and there is a useful section (Section 2) that provides the preliminaries to the material which is relevant and useful. There are some few minor typographical errors in the manuscript. The following suggestions which would improve the quality of the paper should be considered before publication:

1. The authors state that the incorporation of a fractional time-order calculus yields non local and full-memory behavior. This is due to the definition of a fractional derivative involving a convolution integral which is something that the authors should emphasise in the introduction. 2. The authors state that Eqs.(2) is obtained by replacing the integer derivatives with the fractional ones, where αj ∈ (0, 1], this statement is not correct: for example if αj = 1, Eqs.(2) becomes a 2-order dierential equation, meanwhile Eqs.(1) is a 4-order dierential equation. 3. The authors state that It is not hard to see that Eqs.(3) can decline into the Dirichlet boundary value problem Eqs.(2)" wich is not correct, as Eqs.(2) has both left and right Riemann-Liouville derivative. 4. Before Section 3. It is important to concisely explain to the reader why the Sturm-Liouville operator is taken that way, what happen if we permute both the RL derivative and the Caputo derivative for example? will the analysis remain the same?

5. There is no trace study in the work, how sure are we that the terms tD γj−1 T (kj (0)Φp( C 0 D γj t zj (0))) and tD γj−1 T (kj (T)Φp( C 0 D γj t zj (T))) exist? 6. In the 6th to the last line of page 5, it is stated that F(Zk(t)) < − Pj=m j=1 (L ∗ j ) p for all k > ˜k; It is not clear why. that statement needs to be improved. 7. The results was obtained while making quite a lot of hypothesis (H0, H1H2). What happen if we reduce the amount of assumptions? what impact is it going to have on the results? 8. Under Equation (27), you said "Let τ = r Wc ". Did you mean τ = 1 Wc ? 9. In the proof of Theorem 3.2, after "Choose τ ∈ (0, 1 Wc )", it is not clear how the following equations is obtained from Equation (29). 10. The rst line on page 8 should be .... "and F(Z) ≥ 1 pmp kZk p H"... 11. At the second line of Equation (30), ζ p 0 kzk,jk p (γj ,p) should be replaced by |zk,j (t)| p . 12. In the 8th line of page 8, it is stated that 0 ≤ F(Zk(t)) < − Pj=m j=1 kzk,jk p (γj ),p as k → ∞; It is not clear why. I think that it is not correct as stated and can be improved. 13. They concluded with a numerical analysis which seems not to be extensive enough. They should be enhanced to reiterate the primary results given, especially in terms of the Theorems presented. 

Reviewer 3 Report

Report on the article “ Study on infinitely many solutions for a class of Fredholm fractional integro-differential system

 I have read the work. Below you can find my main comments.

In this work, authors study a class of nonlinear fractional Sturm-Liouville boundary value problemsestablishing some sufficient conditions for the existence of solutions. An illustrative example is also presented.

* Check equation (5)

* Who is alpha in Lemma 2.4?

*The physics behind needs to be addressed or at least mentioned. After all, we need to understand the physics behind fractional shaking.

*A discussion section most be included.

Round 2

Reviewer 2 Report

The authors have addressed the questions raised in my original assessment. They have also improved the redaction of the paper. Therefore, the manuscript is suitable for publication.
